# Assessing the Welfare of Technicians during Transits to Offshore Wind Farms

**Tobenna D. Uzuegbunam** [1,*]**, Rodney Forster** [1] **and Terry Williams** [2]

1. Department of Biological and Marine Sciences, University of Hull, Hull HU6 7RX, UK; r.forster@hull.ac.uk
2. Risk Institute, University of Hull, Hull HU6 7RX, UK; terry.williams@hull.ac.uk
* Correspondence: tobenna.uzuegbunam@landuse.co.uk

**Abstract:** Available decision-support tools rarely account for the welfare of technicians in maintenance scheduling for offshore wind farms. This creates uncertainties, especially since current operational limits might make a wind farm accessible but the vibrations from transits might be unacceptable to technicians. We explore technician exposure to vibration in transit based on the levels of discomfort and the likelihood of seasickness occurring on crew transfer vessels (CTVs). Vessel motion monitoring systems deployed on CTVs operating in the North Sea and sea-state data are used in a machine learning (ML) process to model the welfare of technicians based on operational limits applied to modelled proxy variables including composite weighted RMS acceleration (aWRMS) and motion sickness incidence (MSI). The model results revealed poor to moderate performance in predicting the proxies based on selected model evaluation criteria, raising the possibility of more data and relevant variables being needed to improve model performance. Therefore, this research presents a framework for an ML approach towards accounting for the wellbeing of technicians in sailing decisions once the highlighted limitations can be addressed.

**Keywords:** human factors; offshore windfarm; operations and maintenance; whole-body acceleration; welfare assessment; seasickness; comfort

## 1. Introduction

The operations and maintenance (O&M) phase of an offshore wind farm can amount to about a third of the overall life-cycle costs [1]. Efficient maintenance strategies play a key role in reducing the costs and risks associated with the O&M phase [2]. For most offshore wind farms, maintenance activities are usually carried out using crew transfer vessels (CTVs), which transport crew members, spare parts, and technicians to perform various maintenance activities. The planning process for these maintenance activities usually involves a criteria-based decision-making process which is strongly dependent on weather and sea-state conditions [3]. An upper limit of significant wave height of 1.5 m usually applies for CTV operations, as well as the availability of maintenance resources such as access vessels, spare parts, technicians, and operational and regulatory guidelines [4]. As such, decisions are taken on whether transits should be undertaken based on the factors outlined, and these decisions are sometimes aided using decision-support tools [1].

For technicians onboard CTVs in transit, the main concern is their wellbeing and their ability to conduct work upon arrival at offshore wind turbines [5]. However, available maintenance strategies and decision-support models rarely account for the welfare of technicians in the decision-making process [1]. This is important as the literature on the human response to vessel motions suggests that vibrations caused by transits on marine vessels can affect the comfort, health, and ability to work of passengers.

### 1.1. The Effect of Vessel Motions on Technician Welfare

The effects of vibrations on humans, particularly in relation to seafarers, have been explored to include health effects and the effects on comfort and performance [6,7]. The

most common health-based effect of long-term exposure to whole-body accelerations is lower back pain [6,8–10], while the most cited short-term health-based effect is motion sickness [11], which research shows can even affect experienced offshore wind technicians [12].

The general understanding of motion sickness is the neural mismatch theory presented by [13]. The available literature suggests that increasing magnitudes of vertical acceleration and frequencies around 0.2 Hz are responsible for inducing motion sickness [14]; however, later publications including [15,16] have explored the impact of axes other than the z-axis in causing seasickness, suggesting that this phenomenon may not just involve a vertical mismatch. Past research has also associated motion sickness with nausea and vomiting, but the modern literature highlights more symptoms including fatigue, sweating, and reduced cognitive function [17–19]. Therefore, it can be concluded that motion sickness is a complex phenomenon and knowledge about its mechanisms and symptoms is continuously evolving.

Vessel motions can also affect the comfort of passengers. There exists a range of accelerations known to cause discomfort in humans which can lead to distractions and annoyance [20], can affect human reaction time [21], and can cause fatigue in high-speed vessels, as expressed by [22,23]. The full range of these short-term effects has not been fully explored and is not stated in the available standards.

### 1.2. International Standards

International standards provide the most commonly adopted methods for assessing comfort and motion sickness in passengers including the international standard ISO 2631-1 [24], which is the most recent standard for whole-body exposure to vibration. The standards present ways of assessing comfort and seasickness using specific frequency weighting functions which are designed to model, using mathematical digital signal processing, the response of the human body to wave phenomena or model human responses to accelerations based on the axis of acceleration and different postures [24]. This standard presents a way of assessing comfort and motion sickness using the root-mean-square (RMS) of whole-body accelerations by integrating a weighted acceleration squared over a dose period [24], which is the most widely used method for continuous acceleration environments. The equation below presents an expression for the weighted root-mean-square of acceleration.

$$a_w = \left[ \frac{1}{T} \int_0^T a_w^2(t)dt \right]^{\frac{1}{2}} \tag{1}$$

where $a_w(t)$ is the weighted acceleration as a function of time measured in metres per second squared (m/s$^2$) or, for rotational acceleration, measured in radians per second squared (rad/s$^2$), while t is the measurement duration in seconds for an exposure range up to 6 h.

To account for multiple acceleration axes, a vector sum combination can be performed using the expression:

$$a_{xyz} = \sqrt{k_x a_{wx}^2 + k_y a_{wy}^2 + k_z a_{wz}^2} \tag{2}$$

where $a_{xyz}$ is the weighted frequency RMS vector sum of accelerations, and $a_{wx}$, $a_{wy}$, and $a_{wz}$ are the weighted frequency acceleration in the x, y, and z-axis, respectively, while $K_x$, $K_y$, and $K_z$ are multiplying or scaling factors of 1.4, 1.4, and 1.0 for health assessments and 1.0, 1.0, and 1.0 for comfort assessments, respectively [6].

To assess comfort, magnitudes of RMS accelerations are typically used to describe levels of discomfort in passengers [6]. Thresholds of acceptable vibrations have been identified from experimental studies where levels of discomfort have been associated with magnitudes of RMS accelerations, including [25,26]. These subjective scaling mechanisms are presented in ISO 2631-1 and shown in Table 1 below.

**Table 1.** Estimations of comfort response to vibrations (ISO 2631-1, 1997).

| The Magnitude of Acceleration in ms$^{-2}$ | Comfort Reaction |
|---|---|
| Less than 0.315 | Not uncomfortable |
| 0.315–0.63 | A little uncomfortable |
| 0.5–1 | Fairly uncomfortable |
| 0.8–1.6 | Uncomfortable |
| 1.25–2.5 | Very uncomfortable |
| Greater than 2 | Extremely uncomfortable |

Table 1 presents the thresholds of estimated human responses to magnitudes of RMS accelerations as presented by ISO 2631-1. It should be noted that the expressions in Table 1 are approximate thresholds of human reactions to accelerations.

To assess seasickness, the variable most used in the available literature is the likelihood of a passenger vomiting, termed motion sickness incidence (MSI). This is likely because while vomiting can be measured, other known symptoms of motion sickness are subjective. Percentage values between 20% and 25% are typically used as limits of acceptable conditions [5]. MSI is expressed as:

$$\text{MSI} = \text{K}_m \cdot \left\{ \int_0^T [a_w(t)]^2 dt \right\}^{\frac{1}{2}}, \tag{3}$$

where $\text{K}_m$ is a constant equal to 1/3 in a mixed population of men and women but varies based on the population exposed to motions for up to 6 h. Additionally, $a_w(t)$ is the instantaneous frequency-weighted acceleration in the z-axis direction, and t is the measurement of duration in seconds.

Though able to assess the generalised comfort and likelihood of seasickness in passengers, the application of human exposure to accelerations in the maintenance planning of offshore wind farms is rare. This can prove problematic as the literature suggests that the operational limit of a 1.5 m significant wave height set for CTVs can make an offshore wind turbine accessible, but the exposure to vibrations may be intolerable [7].

This paper presents a novel method of accounting for the welfare of technicians in the sail or not-sail decision-making process associated with maintenance planning by modelling the comfort and health of technicians based on proxies including the composite weighted RMS of acceleration (aWRMS), a term used to define the vector sum of RMS whole-body accelerations, to describe levels of comfort, and motion sickness incidence (MSI) to describe the likelihood of seasickness, using a machine learning approach. Both metrics were chosen as measurable and recognisable metrics for the welfare of technicians based on their relevance in literature and the availability of data. The next section of this paper describes the methodology used to achieve the research objective.

## 2. Materials and Methods

### 2.1. Scope

This research assumed that the technicians onboard participating CTVs were in a seated position during transits. Further assumptions were made regarding the placement and calibration of the accelerometers used to measure vessel acceleration described in the subsection below. This research used proxy variables to define technician welfare concerning the short-term effects of acceleration exposure. As such, long-term effects of acceleration exposure such as lower back pain were not considered. The proxies used represented levels of discomfort and the likelihood of seasickness occurring from transits. Estimations for the discomfort of technicians were represented using the composite weighted RMS of acceleration, which described the vector sum of translational RMS accelerations perceived with time and derived from the mathematical expression in Equation (2) above. Estimations for seasickness were represented using motion sickness incidence (MSI), which represented

the percentage likelihood of technicians vomiting and was derived from the mathematical expression in Equation (3) above.

### 2.2. Data and Instrumentation

This research combined the use of operational sea-state data licenced from the Copernicus Marine Service (marine.copernicus.eu, accessed 22 March 2022), and secondary operational in situ data from vessel motion monitoring systems (VMMS) deployed on 12 participating crew transfer vessels ranging in hull length from 18 m to 24 m. The VMMS measured acceleration data in six degrees of freedom, vessel speed, vessel heading, and GPS location data with time stamps. The VMMS was developed, calibrated, and deployed by BMO Offshore [27], a data solution company delivering marine-based operational information and decision support systems, and was made available to this research from the 'Safety and Productivity of Offshore Wind Technician Transit' (SPOWTT), project which was aimed at improving the safety and productivity of offshore turbine technicians [28]. The data collection process in this research commenced in January 2019 and ended in October of the same year, covering eight months, and resulting in eight hundred and fifty ($nt$ = 850) defined O&M transit days after data processing and cleaning. This research defined an O&M transit as a transit originating from an exit port, proceeding to a wind farm, and returning to its or another port of exit.

Operational data products from the Copernicus Marine Service consisted of hindcast datasets for:

I.   Tri-hourly wave data through the Atlantic—European Northwest Shelf product (NWSHELF_REANALYSIS_WAV_004_015), resampled at a daily resolution and provided at approximately 1.5 km resolution from the WAVEWATCH III wave model [29]. The product outputs included wave parameters for the significant wave height, wave period, and directional characteristics.

II.  Daily sea surface height and current hindcast data through the Atlantic—European Northwest Shelf product (NORTHWESTSHELF_ANALYSIS_FORECAST_PHY _004_013), provided at 1.5 km resolution from the NEMO (Nucleus for European Modelling of the Ocean) ocean model [30]. The product provided outputs for current speed, current direction, and sea surface heights.

III. Daily hindcast remotely sensed surface winds from the Global Ocean Wind Product (WIND_GLO_WIND_L4_REP_OBSERVATIONS_012_006). The product provided outputs from scatterometers and radiometers for directional wind velocities and stresses.

All products were expressed between January and October to match VMMS measurements and aggregated by mean to daily mean resolution in order to create daily dose values for technician welfare following the scope of the project.

### 2.3. Study Area

The study area covered latitudes between 51° N and 59° N in the North Sea. This was because VMMS data were collected from vessels deployed in four different wind farms in the North Sea which were operated by four different wind farm operators for spatial variability. The plot in Figure 1 presents a map of the North Sea area, showing operational offshore wind farms in the United Kingdom and the relevant available buoys and met stations considered in this project for metocean data validation. While the map shows operational wind farms in the region, the specific wind farms used to achieve the research aims are not highlighted, following the terms according to which the project data were made available.

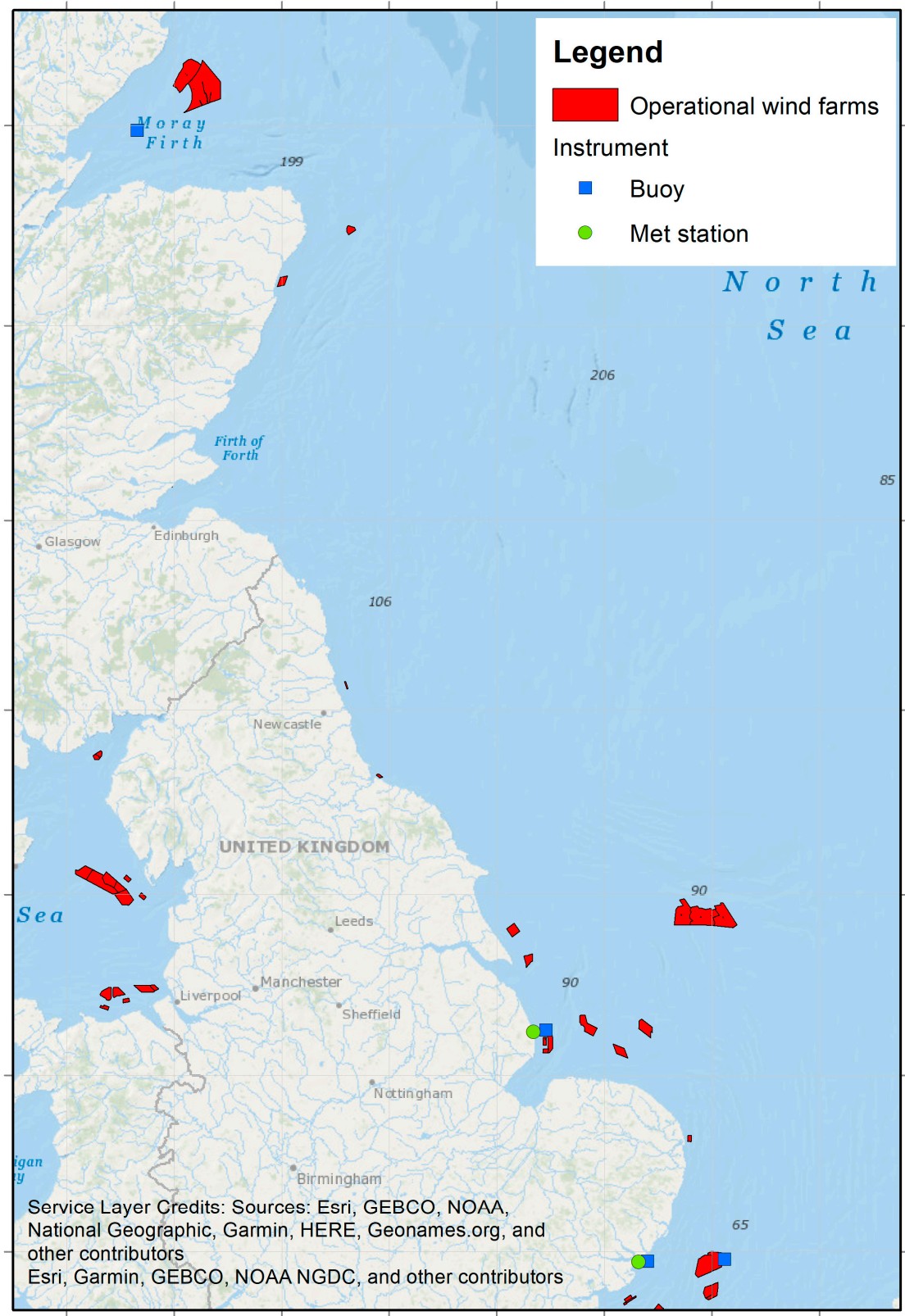

**Figure 1.** Locations of available operational wind farms in red-coloured polygons, with relevant buoys in blue, and met stations in green. The image contains data provided by The Crown Estate that are protected by copyright and database rights, Cefas, licensed under the Cefas WaveNet Non-Commercial Licence v1.0, and the channel coastal observatory (CCO), licenced under the Open Government Licence v3.0.

### 2.4. Modelling the Welfare of Technicians

The objective of this research was to develop a dual-criterion welfare model that aids sailing decisions using proxy variables of the composite weighted RMS acceleration (aWRMS) and motion sickness incidence (MSI) to represent the levels of comfort and the likelihood of seasickness of technicians, respectively. Figure 2 presents the project workflow for achieving the research aim.

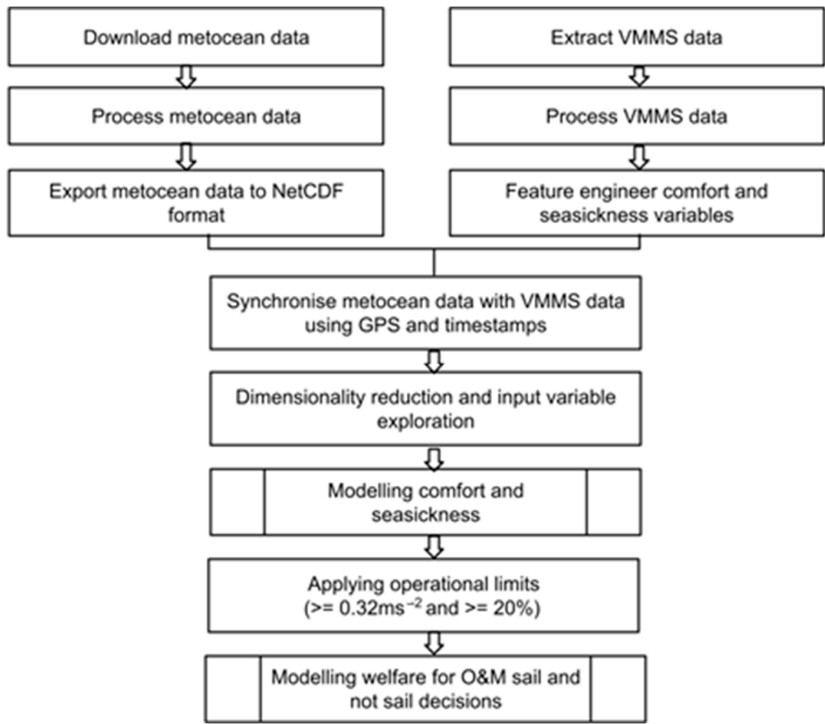

**Figure 2.** Project workflow for assessing the welfare of technicians in transit.

Our first target was to validate the metocean data and ensure that the sea-state data provided sufficient spatial coverage of transit routes travelled by participating CTVs. As such, data from buoys and met masts were used for validation and selected for their positions either in the path of the port of exit for participating CTVs or at the participating wind farms. The data from the in situ instruments were open source data provided by Cefas and funded by Environment Agency, licenced under the Cefas WaveNet Non-Commercial Licence v1.0, and from the channel coastal observatory (CCO) licenced under the Open Government Licence v3.0.

To explore human exposure to accelerations, the meteorological dataset, including variables of significant wave height, wave direction, wave period, sea surface height, current speed, current direction, wind speed, and wind direction, was merged into a dataset using dates as a merging variable in a MATLAB2018a workspace. An additional tidal range variable was created from the 15 min instantaneous resolution of the sea surface height variable in the current data rather than the daily mean to explore the impact of lunar tides on the proxy variables. This was achieved by finding the difference between the daily maximum and minimum sea surface height expressed in metres.

The proxy variables were developed from VMMS acceleration measurements. A variable for aWRMS was created from the expression in Equation (2) using the vector sum of accelerations including the x, y, and z axis, as well as the roll, pitch, and yaw accelerations following ISO 2631-1 guidance. The MSI variable was created from the expression in Equation (3) using the z-axis acceleration, also following ISO 2631-1 guidance. Applied weightings depend on the posture of passengers; as such, weightings in this study were applied under the assumption that technicians on the CTVs are in seated positions. As such, whole-body acceleration weighting for persons in seated positions

was expressed as $W_k$ for z-axis accelerations, $W_d$ for x- and y-axis accelerations, $W_e$ for rotational accelerations, and $W_f$ for z-axis accelerations for MSI, including scaling factors of '$k_x = 1.4$, $k_y = 1.4$, $k_z = 1$' for the MSI and '$k_x = 1.0$, $k_y = 1.0$, $k_z = 1.0$' for aWRMS. The dataset of meteorological variables was then merged with the dataset of VMMS using join functions and dates as the key merging variables. Following this, a dimensionality reduction process was used to select a subset of the most relevant variables for the predictive modelling. This process was conducted to train models faster, simplify the models, improve accuracy, and reduce over-fitting where a model did not generalise well on unseen data based on the training data. As such, this process identified which input variables to include and which irrelevant variables to exclude for predictive modelling. This study used a filtering method to rank variables based on their correlation coefficient univariate metric [31], and principal component analysis (PCA) to identify the most relevant variables that make up most of the variance in predicting the proxy variables. These methods were chosen over the wrapper and embedding [32], which are embedded in the machine learning process and are not model-agnostic [33]. As such, domain knowledge could be applied to the dataset, such as the removal of redundant proxy variables such as the motion sickness dose value (MSDV), which also describes the likelihood of seasickness [34]. Seven variables were identified, accounting for 70% of the variation in predicting both proxy variables including vessel transit duration, vessel speed, vessel heading, significant wave height, current speed, current direction, and tidal height.

For the modelling process, the dataset of the identified variables was split into a training set of 637 transits (75% of the dataset) and a testing set of 212 transits (25%) using a hold-out function. This was performed following a standard machine learning approach to validate the performance of a model selected [35]. Therefore, the machine learning model was defined as the following general equation,

$$\hat{y}_i = \beta_0 + \sum_{i=1}^{n} \beta_i X_i + e_i \tag{4}$$

where $i = 1, 2 \ldots$ n; $\hat{y}_i$ is the proxy variable in the ith sample; $X_i$ comprises the input variables in the ith sample; $\beta_i$ is the coefficient for the input variables and $e_i$ is the residual error.

The training set was trained against multiple regression models including linear regression models, regression trees, support vector machines, Gaussian process regression models, and ensemble trees. For each iteration, we assessed the model's performance by calculating the coefficient of determination ($R^2$) to account for the goodness of fit, and the mean squared error (MSE) and root mean squared error (RMSE) to provide a measure of how far apart model predictions were from estimated values. The machine learning process identified a Gaussian process regression (GPR) model as the model of best fit, which was tested on the testing set using a 'ftrgp' function to predict the proxy variables.

To deliver sailing decisions, we applied defined operational limits to the model's predicting the proxy variables. Operational limits were defined by ISO 2631-1 for limits of human operation and limits of operation based on best seafaring practices. These included a limit equal to or greater than 0.32 ms$^{-2}$ for predicted values of aWRMS discretised as progressively uncomfortable for technicians, and limits applied to the predictions of MSI equal to or greater than 20% discretised as unfavourable sailing conditions able to progressively induce seasickness in 20% or more of the population of technicians in transit. The resulting model was a binary decision support model that predicted sail and not-sail decisions from 'if and else' statements written in code in the MATLAB workspace.

## 3. Results and Discussion

Our first step was to ensure the accurate description of transit routes in project sites and validate metocean data spatial coverage. Data from in situ devices such as buoys and data from met masts were compared with the metocean data at specific points during the transit of participating crew transfer vessels between ports and wind farms. The variables

tested included significant wave height (m) and wind speed (ms$^{-2}$). Table 2 below presents the results of the comparisons with available in situ devices.

**Table 2.** Comparisons between metocean data and in situ measurements showing statistical relevance.

| Category | R$^2$ | RMSE | *p*-Value | Locations |
|---|---|---|---|---|
| Significant wave height | 0.89 | 0.20 | <0.05 | Project site 1 |
| Significant wave height | 0.84 | 0.25 | <0.05 | Project site 3 |
| Significant wave height | 0.86 | 0.15 | <0.05 | Project site 4 |
| Wind speed | 0.54 | 1.74 | <0.05 | Project site 1 |
| Wind speed | 0.847 | 0.213 | <0.05 | Project site 4 |

The comparisons between metocean data and in situ data revealed good relationships between datasets, suggesting that metocean data products were able to provide accurate spatial coverage of the CTV routes when compared with available in situ data. Better relationships were found in the comparisons with significant wave height than with sea surface height and wind speed. This is likely due to the significant amount of missing data points in wind speed data.

*Welfare Modelling*

The process of dimension reduction revealed the most relevant variables to predict aWRMS and MSI, including vessel journey duration, vessel speed, tidal range, current speed, significant wave height, and current direction. Some identified variables are typically present in the parameters used when exploring comfort or the incidence of motion sickness, including journey duration due to the negative relationship between duration and magnitudes of acceleration [36,37], the relationship between vessel speed and magnitudes of acceleration [38], and significant wave height and MSI [34,39–41]. While other variables such as current speed, current direction, and tidal range are not typically present in parameters used in studies evaluating comfort and MSI, the dimensionality reduction process identified these parameters as relevant to the variance in predicting aWRMS and MSI. The identified variables were used as input variables to identify the model of best fit by training the training set against multiple regression models, shown in Table 3, which revealed a rational quadratic regression model.

**Table 3.** Summary of trained models used to identify the model of best fit.

| Regression Model | R$^2$ (aWRMS) | RMSE (aWRMS) | R$^2$ (MSI) | RMSE (MSI) |
|---|---|---|---|---|
| Linear | 0.51 | 0.08 | 0.29 | 4.63 |
| Interactions linear | 0.52 | 0.08 | 0.27 | 4.68 |
| Robust linear | 0.50 | 0.09 | 0.28 | 4.64 |
| Stepwise linear | 0.53 | 0.08 | 0.27 | 4.69 |
| Fine tree | 0.40 | 0.09 | 0.25 | 4.77 |
| Medium tree | 0.43 | 0.09 | 0.30 | 4.62 |
| Coarse tree | 0.46 | 0.09 | 0.31 | 4.57 |
| Linear SVM | 0.50 | 0.09 | 0.27 | 4.68 |
| Quadratic SVM | 0.54 | 0.08 | 0.37 | 4.35 |
| Cubic SVM | 0.54 | 0.08 | 0.38 | 4.32 |
| Fine Gaussian SVM | 0.21 | 0.12 | 0.24 | 4.78 |
| Medium Gaussian SVM | 0.56 | 0.08 | 0.41 | 4.20 |
| Coarse Gaussian SVM | 0.51 | 0.08 | 0.33 | 4.47 |
| Boosted tree | 0.58 | 0.08 | 0.45 | 4.02 |
| Bagged trees | 0.56 | 0.08 | 0.45 | 4.02 |
| Squared exponential GPR | 0.61 | 0.08 | 0.43 | 4.02 |
| Matern 5/2 GPR | 0.59 | 0.08 | 0.42 | 4.02 |
| Exponential GPR | 0.60 | 0.08 | 0.43 | 4.02 |
| **Rational quadratic GPR** | **0.63** | **0.07** | **0.46** | **4.02** |

The Gaussian process regression (GPR) model is a non-parametric method for machine learning regression which was used on a testing set. We show the results of the model generated using the MATLAB 'ftrgp' function with 'KernelFunction = rationalquadratic', 'BasisFunction = constant', 'Standardize = true', 'PredictMethod = Exact', as the model specification. The accuracy of the models was measured using their $R^2$, MSE, and RMSE values, which are best used in measuring regression model performance [42]. These measures were chosen to identify how well the models predicted the proxy variables and present measures through which improved models can be tested against. Figures 3 and 4 present the results of both models in predicting proxy variables.

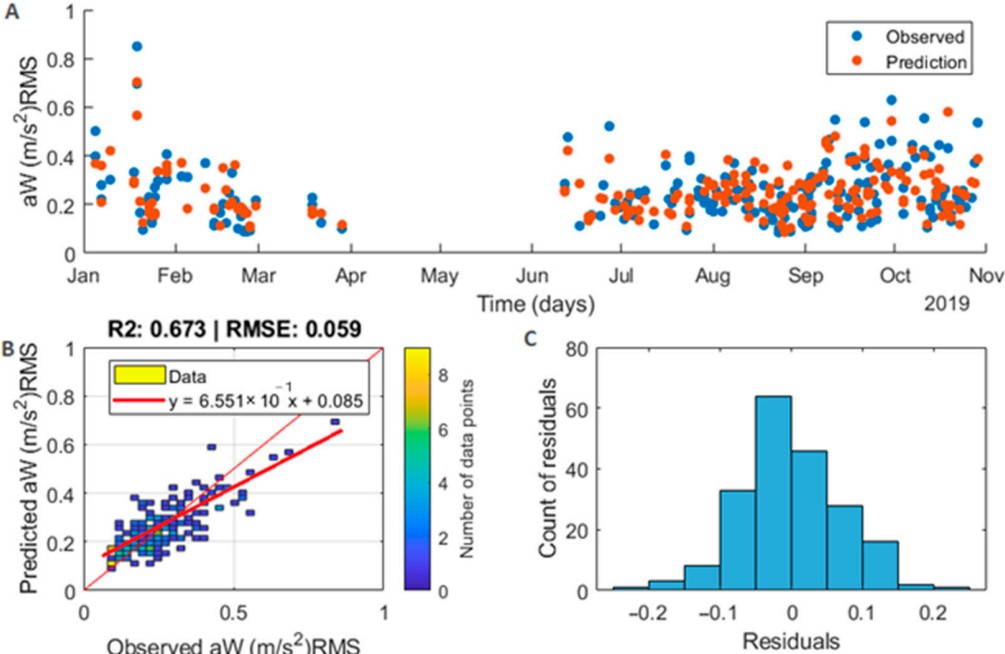

**Figure 3.** Plots showing the model performance. (**A**) A response plot showing observed (measured), in blue, and predicted composite weighted RMS acceleration (aWRMS), in orange, with time. (**B**) The correlation between predicted and observed aWRMS (m/s$^2$) showing the density of data points. (**C**) A histogram of residuals.

Figures 3 and 4 present model results for predicting the proxy variables. The response plot in (A) of both figures shows that predictions follow similar tracks to measured values. The plot of the correlation between predicted and measured aWRMS (Figure 3B) shows a moderate relationship between predicted and estimated values with an $R^2$ value of 0.67. This shows that for this model, 67% of the variation of estimated aWRMS can be explained by the independent variables, and as such, only 33% reside in the residual. This provides a measure of improvements that need to be implemented in improving the model, which could include less significant measures such as increasing the size of the dataset to improve predictability and more significant measures such as unexplored or unknown factors that could influence the measured acceleration on the participating CTVs. The model's errors are evaluated from RMSE values of 0.06 ms$^{-2}$, and the histogram of residuals (Figure 3C) which shows that model predictions can be off by 0.25 ms$^{-2}$; however, most data points are centred around zero, which suggests that the errors are not significant. Additionally, the presence of data points (the density plot in Figure 3B) above the perfect prediction line (thin red line) between 0.1 and 0.3 ms$^{-2}$ and data points below the prediction line between 0.5 and 1.0 ms$^{-2}$ shows that the model overestimated more common values of composite weighted acceleration and underestimates the less common values. Further statistical significance is presented by a *p*-value less than 0.05.

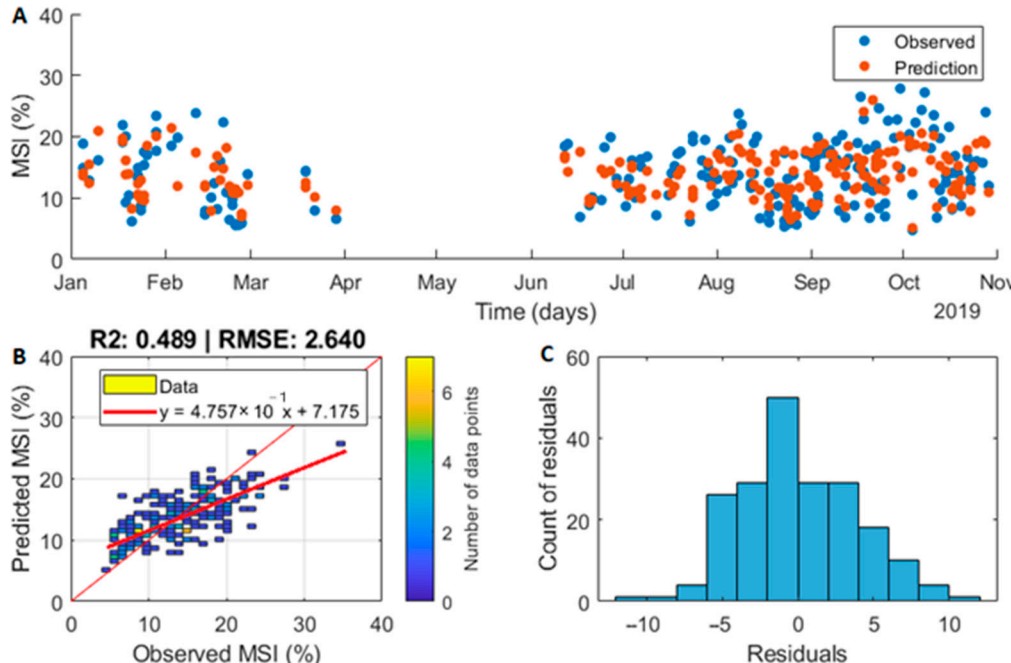

**Figure 4.** Plots showing the model performance. (**A**) A response plot showing observed (measured), blue, and predicted motion sickness incidence (MSI), orange, with time. (**B**) The correlation between predicted and observed MSI (%) with a density of data points. (**C**) A histogram of residuals.

Similarly, Figure 4B presents a poor relationship between the predicted and estimated values of motion sickness incidence (MSI), with an $R^2$ value of 0.49. This shows more than half of the variance, up to 51%, resides in the residual. As such, the variables explored do not make up most of the variation in predicting MSI. This shows that a significant amount of work needs to be done to improve the models' prediction, and as such, the model does not present an acceptable level of performance. The RMSE values of 2.6% and the histogram of residuals in Figure 4C show that model predictions can be off by 2.6%; however, a *p*-value less than 0.05 shows that there is statistical significance. Table 4 presents the results of both models between training and testing sets.

**Table 4.** Summary of model results for training and testing datasets.

|  | RMSE | $R^2$ | MSE | Speed (s) | Time (s) | Model |
|---|---|---|---|---|---|---|
| Training set | 0.07 | 0.63 | 0.005 | 25,000 | 12.11 | aWRMS Rational |
| Testing set | 0.06 | 0.67 | 0.004 | 8800 | 3.05 | quadratic GPR (ms$^{-2}$) |
| Training set | 4.02 | 0.46 | 16.15 | 24,000 | 16.56 | MSI Rational quadratic |
| Testing set | 2.64 | 0.49 | 6.97 | 8800 | 12.11 | GPR (%) |

It should be noted that the models do not predict comfort and seasickness in technicians, but rather the aWRMS and MSI as proxies for the levels of discomfort using the magnitude of aWRMS and the likelihood of seasickness using the percentage value of MSI. In addition, the comfort thresholds presented in the ISO 2631-1 are based on dated experimental studies [26] which might not be suitable for current assessments or suited to measurements on CTVs. Furthermore, MSI, as stated in Section 1, is the incidence of vomiting (a symptom of motion sickness) and does not account for other symptoms of motion sickness including sweating, changes in temperature, headaches [28], or other susceptibility factors associated with motion sickness such as temperature and a lack of visual reference [43]. This is highlighted in a recent publication by [43], who suggested improvements to the international standard to include more dimensions of motion sickness. As such, motion sickness incidence may not be a sufficient indicator of seasickness.

To predict sail and not-sail decisions, operational limits are applied to the model outputs of the aWRMS and MSI based on the operational limits defined by ISO 2631-1 for providing an approximate human comfort response to accelerations, and the best seafaring practices for limiting the incidence of motion sickness [5,8]. The limits applied include the categorisation of increasing values of RMS acceleration above 0.315 ms$^{-2}$ as progressively uncomfortable and MSI values below 20% as limits of acceptable conditions. Therefore, the model created is based on a dual-criterion system established from magnitudes of predicted weighted acceleration and the predicted incidence of vomiting. Figure 5 presents the results of the technician welfare model with predicted sail and not sail decisions over three months.

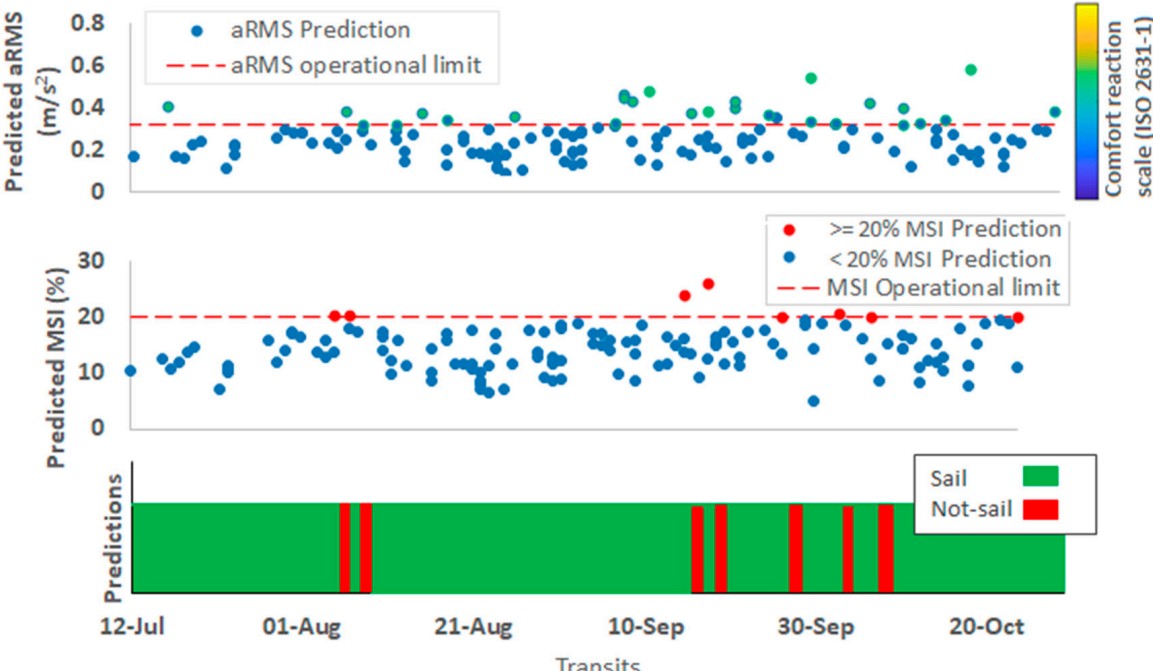

**Figure 5.** A plot of the technician welfare model showing sail and not-sail decisions based on applied operational limits.

Figure 5 presents 159 predictions based on the dual criterion operational limits applied to predictions of aWRMS and MSI. The coloured balls represent model predictions. For the aWRMS model, the balls range between blue and green showing levels of discomfort based on ISO 2631, and for the MSI model, the red balls indicate greater than or equal to 20% MSI values while the blue balls indicate less than 20% MSI. The final well-being model shows green, for sail, and red, for not-sail, bars which indicate sailing decisions derived from aWRMS and MSI model outputs. The welfare model predicted 129 sail decisions (shown in green) and 7 not-sail decisions (in red).

## 4. Conclusions

This research presents a holistic approach to an aspect of operations and maintenance planning typically not expressed in accounting for the welfare of technicians in transit by assessing comfort and the likelihood of seasickness. This can further inform their ability to perform work upon arrival at offshore turbines. Proxy indicators were used to describe comfort levels and the likelihood of seasickness in technicians, including aWRMS and MSI, respectively. These proxies used were selected due to their relationships with whole-body accelerations, which are supported by findings in the literature, including the relationship between the magnitude of RMS accelerations and levels of discomfort [6] and MSI as an indicator of the incidence of seasickness [5]. Significant contributions to the literature were made in describing the non-homogeneous random variables of sea-state and vessel

operational parameters, as typically, studies on human exposure to acceleration calculate the RMS of vessel acceleration using a response amplitude operator (RAO) along with wave energy spectra to produce response spectra [42] or combine numerical ocean models to seat models, providing a range of sea state and vessel parameters for analysis [22]. This is because experimental in situ measurements of sea-state and vessel characteristics have limited spatial coverage and are time-consuming and expensive [22,44]. This research, however, benefits from measured vessel accelerations which allowed more variable exploration of parameters to identify relevant relationships for modelling.

The results of the models created present a framework for assessing welfare based on proxy variables; however, the performance of the models indicated that more work is needed to improve model predictability. In application, the welfare of technicians can be assessed before or alongside a maintenance transfer plan able to make sailing decisions that can account for the wellbeing of technicians. Therefore, this research can provide major contributions to the maintenance planning for offshore wind farms once the major limitations, outlined in Section 4.1, have been addressed. This is particularly important as typical CTV operational limits can make a wind farm accessible but perceived accelerations from transits may be unacceptable to technicians [7], potentially causing discomfort and seasickness [6] which can affect the ability to perform complex functions in technicians, including manual handling and cognitive tasks [18]. This research can also provide a framework for regulatory compliance, as the model-set acceleration limit falls below the limits set in the Control of Vibration at Work Regulations 2005 [45] which came into force in the UK in 2005, and the limits to MSI also aid wind farm operators with their duties of care to technicians under the Health and Safety at Work etc Act 1974 [46].

### 4.1. Limitations and Future Work

There are key limitations in this research that need to be addressed with areas where future work can be done.

The VMMS data used in this paper were collected for a different purpose than this project's objectives and from a project with different objectives. The authors of this paper had limited control over the measurement of acceleration data, the positioning of the VMMS devices, and the calibration of the devices. In addition, there were restrictions on the data made available including the vessels used and subjective measurements, which prevented the validation of model results. For this reason, some assumptions had to be made within the course of this research regarding the calibration of the accelerometers in the VMMS and the placement of the VMMS on the participating vessels, which may have created errors due to the physical constraints of the device and the location of the device in relation to the personnel on board. The assumptions were made based on informal requests to the researchers within the original study, as well as the publication from the study [28]. In addition to this, the research objective had to be amended to the dataset available, which restricted the type of research questions generated for the project. Future research in this field will require a dataset designed for the research objectives to eliminate restrictions in data analysis and exploration, and reduce random errors.

The results of the model-predicted proxies could benefit from validations against real-life measurements of the comfort and seasickness of technicians in transit in the form of measured physiological changes or symptomatic questionnaires [43]. This would also inform the future exploration of input parameters and greatly improve the model. In addition to this, experimental studies cited in the international standard explored limits of operation for vessels of varying sizes, some of which are different from the CTVs used in O&M activities. As such, the operational limits can also be validated against measurements of comfort for CTVs.

Consideration also needs to be performed for the proxies used, including how discomfort is described and the definition of seasickness. While the relationship between RMS acceleration and the discomfort of passengers has been explored in the literature, the available literature is dated, and the nature of questionnaires used to assess discomfort and

the type of vessels or simulations used can impact the ranges of discomfort experienced. Similarly, the model predicts the incidence of vomiting rather than the incidence of motion sickness and does not account for other relevant factors or other susceptibility factors associated with motion sickness [43]. For the technicians onboard CTVs completing complex work, the point of vomiting is well past the point where complex work can be performed, and as such, MSI may not be a useful benchmark for predicting wellbeing.

Future work also needs to be performed to iteratively improve model predictions by exploring more relevant variables for predicting aWRMS and MSI, including increasing the size of the dataset used in making predictions, applying other measurable welfare parameters not included in the welfare model, increasing the spatial variability to account for the sea-state characteristics of other regions, and improving vessel variability in predictions such as different sizes of CTVs, and other service vessels such as service operation vessels (SOVs) and daughter crafts.

**Author Contributions:** Conceptualisation, T.D.U.; methodology, T.D.U.; software, T.D.U.; resources, T.W.; data curation, T.D.U.; writing—original draft preparation, T.D.U.; writing—review and editing, R.F. and T.W.; visualisation, T.D.U.; supervision, R.F. and T.W. All authors have read and agreed to the published version of the manuscript."

**Funding:** This research was supported by funding provided to the lead author by The University of Hull.

**Data Availability Statement:** Sea-state data for this study were obtained using Copernicus Marine Service products. Product information is presented below. Copernicus Marine Service Information, n.d. Atlantic- European Northwest Shelf- Wave Physics Reanalysis, NWSHELF_REANALYSIS _WAV_004_015. [Product] marine.copernicus.eu, https://doi.org/10.48670/moi-00060 [accessed on 20 January 2022]. Copernicus Marine Service Information, n.d. Atlantic—European Northwest Shelf—Ocean Physics Analysis and Forecast, NORTHWESTSHELF_ANALYSIS_FORECAST_PHY_004_013 [Product] marine.copernicus.eu, https://doi.org/10.48670/moi-00054 [accessed on 20 January 2022]. Copernicus Marine Service Information, n.d. Global Ocean Wind L4 Reprocessed 6 hourly Observations, WIND_GLO_WIND_L4_REP_OBSERVATIONS_012_006. [Product] marine.copernicus.eu, https://doi.org/10.48670/moi-00185 [accessed on 20 January 2022]. The authors elect to not share specific vessel data due to personal and commercial confidentiality.

**Acknowledgments:** This research was supported by funding to the first author from the University of Hull. The authors would like to thank BMO Offshore for providing access to VMMS.

**Conflicts of Interest:** The authors declare no conflict of interest.

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
