# Peer review of "Assessing the Welfare of Technicians during Transits to Offshore Wind Farms"

_vibration, doi:10.3390/vibration6020027_

Round 1

Reviewer 1 Report

11) The publication of this work in a scientific journal is in my view premature in view of the severe limitations of the proposed models as evidenced by the weak correlation results obtained (R2 = 0.67 and 0.49). The limitations have been recognized by the authors themselves in section 4.1 and provide interesting avenues for improvement. Although the attempt to relate operational parameters to some welfare proxies is interesting, the poor performance of the models as reported in this paper prevents them from having any significant value for making any reliable sail and no-sail decisions at this time.

22) One major error in this work is computing aWRMS values with scaling factors (kx=1.4, ky=1.4, kz=1) which are erroneously applied for evaluating comfort effects (see clause 8.2 of the ISO 2631-1 standard), thus resulting in values of aWRMS  that are not intended to be compared with the values listed in Table 1 as was done in this paper. The values listed in Table 1 are extracted from the ISO 2631-1 standard and represent the overall vibration total values computed from the root-sum-of-squares of the point vibration total values to assess comfort reactions.  For a seated posture (as presumably was the situation in this work), the multiplying factors should have been  kx=ky=kz=1 and the computation of aWRMS should possibly have included, in addition to translational accelerations on the seat, also measurements of rotational accelerations on the seat and accelerations at the feet and backrest (if any) to justify a comparison with the values given in Table 1.

33) The exact location of the measurements in the vessels is not indicated in the paper but it is specified that six degree-of-freedom measurements were made. Why were the other three degree-of-freedom (presumably rotational accelerations) measurements not included in the computation of aWRMS?

44) In section 4.1 it is stated that some assumptions had to be made regarding the calibration of the accelerometers and their placement on the vessels. If the models developed in this paper were based on measurements that are uncertain, how can we be convinced of their validity?

55) In section 2.4, it is not clear what the reason was for selecting a threshold value of 20% on MSI to set the sailing decisions. Is there a scientific basis for selecting such a value?

66) The inclusion of Figure 1 is not considered to add any significant value to the content of this paper.

77) The models as proposed suppose that the discomfort and incidence of motion sickness entirely rely on operational parameters but omit to consider other factors (ex. individual, organizational) which may also have an important influence on accounting for the wellbeing of technicians. In view of this important limitation, what makes the authors believe that such models have the potential to eventually become reliable tools in the sail or not-sail decision-making process?  

Author Response

Response to Reviewer Comments

The response to comments is written in red starting with an indication of where reference page numbers and line numbers can be found in the revised manuscript.

Point 1: Is the research design appropriate?

Page 6 line 212; lines 217-220

Clarifications and amendments to the research design have been made addressing the scaling factors used, and the vector sum of accelerations performed which includes translational and rotational accelerations (roll, pitch, and yaw).

Point 2: Are the methods adequately described? Are the results clearly presented?

Page 8 lines 279-288; page 11 lines 344-353

Further descriptions of input variables used is provided as well as further explanation of the thresholds applied (including MSI) and the model results (proxy indicators) in predicting technician well-being.

Point 3: The publication of this work in a scientific journal is in my view premature in view of the severe limitations of the proposed models as evidenced by the weak correlation results obtained (R2 = 0.67 and 0.49). The limitations have been recognized by the authors themselves in section 4.1 and provide interesting avenues for improvement. Although the attempt to relate operational parameters to some welfare proxies is interesting, the poor performance of the models as reported in this paper prevents them from having any significant value for making any reliable sail and no-sail decisions at this time.

The model results highlight the need for improvements which are addressed in the study and the lack of validation is also highlighted as a major limitation in the study. This study presents a framework for creating a model from machine learning processes to relate operational parameters to welfare proxies which has not been explored. Therefore, the poor performance of the model is a significant finding in itself and the suggestions for improvements show where further research can be made to improve accuracy and values for sailing decisions.

Point 4: One major error in this work is computing aWRMS values with scaling factors (kx=1.4, ky=1.4, kz=1) which are erroneously applied for evaluating comfort effects (see clause 8.2 of the ISO 2631-1 standard), thus resulting in values of aWRMS that are not intended to be compared with the values listed in Table 1 as was done in this paper. The values listed in Table 1 are extracted from the ISO 2631-1 standard and represent the overall vibration total values computed from the root-sum-of-squares of the point vibration total values to assess comfort reactions.  For a seated posture (as presumably was the situation in this work), the multiplying factors should have been  kx=ky=kz=1 and the computation of aWRMS should possibly have included, in addition to translational accelerations on the seat, also measurements of rotational accelerations on the seat and accelerations at the feet and backrest (if any) to justify a comparison with the values given in Table 1.

Page 6 line 212; lines 217-220

Amendments were made to show that scaling factors used were kx=1.4, ky=1.4, kz=1 from 7.2.3 of the ISO for health assessments (MSI), while the aWRMS model used (kx=1.0, ky=1.0, kz=1.0) from 8.2.2.1 of the ISO, and the vector sum of accelerations performed which includes translational and rotational accelerations (roll, pitch, and yaw).

Point 5: The exact location of the measurements in the vessels is not indicated in the paper but it is specified that six degree-of-freedom measurements were made. Why were the other three degree-of-freedom (presumably rotational accelerations) measurements not included in the computation of aWRMS?

Page 6 line 212; lines 217-220

As stated above, aWRMS values include x, y, z, roll, pitch, and yaw accelerations. A number of iterations were performed in this research with regards to improving model performance including variations of the axis of acceleration applied. Without a validation phase, however, the ISO 2631 was applied using the z-axis for MSI estimations and the vector sum of six-axis for the aWRMS estimations.

Point 6: In section 4.1 it is stated that some assumptions had to be made regarding the calibration of the accelerometers and their placement on the vessels. If the models developed in this paper were based on measurements that are uncertain, how can we be convinced of their validity?

Page 12 lines 406-410 and 414-415

Access is highlighted as the key limitation of this study, and there was limited control over the measurement of acceleration data, the positioning of the VMMS devices, and the calibration of the devices. In addition, there were restrictions on the available data, including the vessels used and subjective measurements, which prevented the validation of model results. It was also understood that some of the restrictions were put in place by the company (BMO) that owns and deployed the VMMS. The assumptions were made from the publication from the study and informal requests to researchers within the originating study where it is understood that the accelerometers are placed under the seats. However, as this is not explicitly detailed, it has been omitted from this manuscript.

Point 7: In section 2.4, it is not clear what the reason was for selecting a threshold value of 20% on MSI to set the sailing decisions. Is there a scientific basis for selecting such a value?

Page 3 line 102

Some clarity has been given to the threshold for MSI is usually based on experience as the likelihood of vomiting in the number of technicians on board, as stated on page 11 lines 353-354. As such, the percentage value aids operators in decision-making.

Point 8: The inclusion of Figure 1 is not considered to add any significant value to the content of this paper.

While data restrictions prevent the wind farms to be highlighted, Figure 1 also shows the locations of in-situ instruments used to validate metocean data and highlights the spatial limitations of in-situ instruments which were important for the methodology employed in this study.

Point 9: The models as proposed suppose that the discomfort and incidence of motion sickness entirely rely on operational parameters but omit to consider other factors (ex. individual, organizational) which may also have an important influence on accounting for the well-being of technicians. In view of this important limitation, what makes the authors believe that such models have the potential to eventually become reliable tools in the sail or not-sail decision-making process?  

The model limitations have been highlighted in section 4.1 including the factors omitted. The study presents a framework and informs on the opportunities that exist in creating a more robust model that includes more factors that should perform better with validation. The relevant factors have been highlighted as well as opportunities to improve model performance to create a reliable tool. It should, however, be noted the individual factors will most likely be missing from future models as they are from the proxy models used as some of these factors are not yet measurable.

Reviewer 2 Report

The aim of this study was to develop a framework to account for the well-being of technicians in sailing decisions for offshore wind farms. The effect of vibration on comfort and motion sickness has been summarised based on recent literature and ISO standard.  Interesting results were presented using the existing vibration and sea condition data to predict comfort and motion sickness. The manuscript is clearly structured and well-written and contains a detailed experimental plan and data analysis.

The limitation of the current standard on comfort and motion sickness prediction has been discussed in recent literature and worth to highlight these limitations on your prediction framework. E.g. current motion sickness MSI was focused on the severe symptom (i.e. vomiting), mild and moderate symptoms may be ignored. Current comfort scaling is based on old literature with lab experiments performed on rigid seats, whereas your technicians may have adopted different postures on different seating arrangements.

The development of the prediction model is heavily linked with the vibration acceleration data. Please give more details on the vibration measurement system, VMMS, including accelerometers, data acquisition system, calibration procedure, measurement location, etc. The author has explained the lack of control of vibration measurement in the limitation section but is still worth giving details on the current used system.

The other important point to account for is the technician’s ability to perform the job after exposure to vibration, e.g. the after-effect of vibration exposure. This is not studied enough and currently no guidance from ISO standards. Could you kindly add some comments on this.

Duration of the journey is also a common contributing factor to discomfort and motion sickness, please discuss how this will affect your prediction model.

Author Response

Response to Reviewer Comments

The response to comments is written in red starting with an indication of where reference page numbers and line numbers can be found in the revised manuscript.

Point 1: Are the methods adequately described?

Page 8 lines 279-288; page 11 lines 344-353

Further descriptions of input variables used are provided and further explanation of the thresholds applied (including MSI) and the model results (proxy indicators) in predicting technician well-being. The additional comments below address the methods' amendments in more detail.

Point 2: The limitation of the current standard on comfort and motion sickness prediction has been discussed in recent literature and worth to highlight these limitations on your prediction framework. E.g., For current motion sickness MSI was focused on the severe symptom (i.e. vomiting), mild and moderate symptoms may be ignored. Current comfort scaling is based on old literature with lab experiments performed on rigid seats, whereas your technicians may have adopted different postures on different seating arrangements.

Page 11 lines 341-350; Page 13 lines 439-445

The authors acknowledge limitations to the international standard highlighted in recent literature and briefly express this in section 4.1. The limitations have been made clearer in the sections highlighted stating the limitations in addressing symptoms and thresholds set from dated literature

Point 3: The development of the prediction model is heavily linked with the vibration acceleration data. Please give more details on the vibration measurement system, VMMS, including accelerometers, data acquisition system, calibration procedure, measurement location, etc. The author has explained the lack of control of vibration measurement in the limitation section but is still worth giving details on the current used system.

Page 12 lines 406-410 and 414-415

Access is highlighted as the key limitation of this study, and there was limited control over the measurement of acceleration data, the positioning of the VMMS devices, and the calibration of the devices. In addition, there were restrictions on the data made available including the vessels used and subjective measurements which prevented validation of model results. The assumptions were made from informal requests to researchers within the originating study as well as the publication from the study where it is understood that the accelerometers are placed under the seats. It was also understood that some of the restrictions were put in place by the company (BMO) that owns and deployed the VMMS.

Point 4: The other important point to account for is the technician’s ability to perform the job after exposure to vibration, e.g. the after-effect of vibration exposure. This is not studied enough and currently no guidance from ISO standards. Could you kindly add some comments on this?

Page 2 lines 63-67

A brief reference is made to the after-effects of exposure such as fatigue and the effect on cognitive function. However, this is not present in the standard and a line has been added to address this. Overall, there are more studies on the long-term after-effects of exposure (back pain mostly) and there seem to be fewer studies addressing short-term after-effects.

Point 5: Duration of the journey is also a common contributing factor to discomfort and motion sickness, please discuss how this will affect your prediction model.

Page 8 lines 280-289

The authors acknowledge the importance of such contributing factors and have expanded on this in Section 3.1 to include the variables identified in the process and the relation to common parameters used in studies exploring human response to motions including the duration of the journey.

Reviewer 3 Report

1. Add the number/location of the approved institutional human subject forms.

2. Indicate the location of the accelerometers on the seats.

3. What types of questionnaires were used in the data collection?

4. The role of rotational motions will be very relevant to discomfort and seasickness in such environments. Please address this issue in the Discussion section.

Author Response

Response to Reviewer Comments

The response to comments is written in red starting with an indication of where reference page numbers and line numbers can be found in the revised manuscript.

Points 1-3:

  1. Add the number/location of the approved institutional human subject forms.
  2. Indicate the location of the accelerometers on the seats.
  3. What types of questionnaires were used in the data collection?

Page 12 lines 406-410 and 414-415

Human subjects were not introduced at this stage of the research but were highlighted as an important means of validating the model results.

Access is highlighted as the key limitation of this study, and there was limited control over the measurement of acceleration data, the positioning of the VMMS devices, and the calibration of the devices. In addition, there were restrictions on the data made available including the vessels used and subjective measurements which prevented validation of model results. The assumptions were made from informal requests to researchers within the originating study as well as the publication from the study where it is understood that the accelerometers are placed under the seats. It was also understood that some of the restrictions were put in place by the company (BMO) that owns and deployed the VMMS.

Point 4: The role of rotational motions will be very relevant to discomfort and seasickness in such environments. Please address this issue in the Discussion section.

Page 6 line 212; lines 217-220

Clarifications and amendments to the research design have been made addressing the scaling factors used, and the vector sum of accelerations performed which includes translational and rotational accelerations (roll, pitch, and yaw).

Reviewer 4 Report

Reviewer Comments to Author(s):

Main comments

The paper presents a study looking at the effect of vibration on workers travelling in boats to offshore wind farms.  The effect of vibration on the discomfort and motion sickness likely to be experience is presented.  The study compares data from a model based on various environmental factors (I think) with the actual measured values.  The computer-based model used was ‘trained’ on data and the predictions are compared with measured data.  These data can be used to estimate the effect on people travelling in the boats.  I offer my comments on the manuscript. 

The page and line numbers below correspond to those on the draft manuscript.

Comments

Page 1, Line 31

Reference ‘Dinwoodie, 2016’ is not included in the references section.

Page 2, Line 88

The multiplication factors used in the study are 1.4, 1.4, 1.0 for x, y, z axes, respectively.  Further it is stated that the data will be used to determine the effect of discomfort among passengers in the boat.  The factors that should have been used to assess discomfort are 1.0, 1.0, 1.0 for x, y, z axes, respectively.  That is, they are unity for the three axes.  The factors used (1.4, 1.4, 1.0 for x, y, z axes) are for determining the effects on health of the passengers.  This will have to be corrected.  What effect will this basic understanding have on the results from the comparison?

Page 3, Line 95

The vibration magnitudes presented in the table should not show any zeros after the decimal point as this could imply the accuracy in the measured data.  Furthermore, the zeros are not included in the table presented in ISO 2631-1 (1997).

Page 4, Line 135

Reference should be made to equation 3 and not equation 2.

Page 6, Line 192

The units for acceleration should be ‘ms-2’ and not ‘ms-1’ (velocity).

Page 7, Line 251

The term ftrgp should be in single inverted commas: ‘ftrgp’.

Page 7, Line 260

Change “… from if and else statements …” to “… from ‘if and else’ statements …”.

Page 9, Line 286

The term MAE is not defined.

Page 9, Lines 291-294

There is confusion over the terms used: observed, calculated, predicted.  Do you mean ‘measured’ rather than ‘observed’?  You also use the term ‘calculated’ for ‘observed’ (see line 292).  But both the measured data and that from the model were ‘calculated’.  This will also apply to Figure 4 on page 10.

Page 9, Line 291

The bottom left figure shows an equation denoted as a red line.  However, the red line drawn is not as that in the equation.  The line drawn corresponds to perfect correlation between the two factors.  Although attempts are made to explain the figures (see lines 313-314), this should be made clear to avoid confusion.  This also applies to Figure 4 on page 10.

Page 10, Line 310

Ensure superscript is used for ‘-2’ in ‘ms-2’.  There are many other occurrences of this.

Author Response

Response to Reviewer Comments

The response to comments is written in red starting with an indication of where reference page numbers and line numbers can be found in the revised manuscript.

Comments

Point 1: Page 1, Line 31

Reference ‘Dinwoodie, 2016’ is not included in the references section.

Page 1 line 30; page 13 line 468-470

The reference has been updated in-line and at the reference list

Point 2: Page 2, Line 88

The multiplication factors used in the study are 1.4, 1.4, 1.0 for x, y, z axes, respectively.  Further it is stated that the data will be used to determine the effect of discomfort among passengers in the boat.  The factors that should have been used to assess discomfort are 1.0, 1.0, 1.0 for x, y, z axes, respectively.  That is, they are unity for the three axes.  The factors used (1.4, 1.4, 1.0 for x, y, z axes) are for determining the effects on the health of the passengers.  This will have to be corrected.  What effect will this basic understanding have on the results from the comparison?

Page 6 line 212; lines 217-220

Amendments were made to show that scaling factors used were kx=1.4, ky=1.4, kz=1 from 7.2.3 of the ISO for health assessments (MSI), while the aWRMS model used (kx=1.0, ky=1.0, kz=1.0) from 8.2.2.1 of the ISO, and the vector sum of accelerations performed which includes translational and rotational accelerations (roll, pitch, and yaw).  

Point 3: Page 3, Line 95

The vibration magnitudes presented in the table should not show any zeros after the decimal point as this could imply the accuracy in the measured data.  Furthermore, the zeros are not included in the table presented in ISO 2631-1 (1997).

Page 3 line 96

The decimal points were added for consistency, however, the implication is understood and has been updated.

Point 4: Page 4, Line 135

Reference should be made to equation 3 and not equation 2.

Page 4, Line 136

Reference corrected from Equation 2 to Equation 3.

Point 5: Page 6, Line 192

The units for acceleration should be ‘ms-2’ and not ‘ms-1’ (velocity).

Page 6, Line 193

The units for acceleration were amended to ‘ms-2

Point 6: Page 7, Line 251

The term ftrgp should be in single inverted commas: ‘ftrgp’.

Page 7, Line 254

The term ftrgp is placed in inverted commas: ‘ftrgp’. Additionally, other functions have also been placed in inverted commas.

Point 7: Page 7, Line 260

Change “… from if and else statements …” to “… from ‘if and else’ statements …”.

 Page 7, Line 263

If and else statements have been placed in inverted commas as ‘if and else’.

Point 8: Page 9, Line 286

The term MAE is not defined.

Page 9, Line 286

RMSE and MAE (Mean Absolute Error) are usually used to quantify how well a model fits a dataset, however, as consideration is made to observation further away from the mean in this study MAE can be a redundant metric.

Point 9: Page 9, Lines 291-294

There is confusion over the terms used: observed, calculated, predicted.  Do you mean ‘measured’ rather than ‘observed’?  You also use the term ‘calculated’ for ‘observed’ (see line 292).  But both the measured data and that from the model were ‘calculated’.  This will also apply to Figure 4 on page 10.

Page 9, Lines 303-311

For consistency and clarity, 'observed' and 'measured' are used not 'calculated' as predicted values are also calculated.

Point 10: Page 9, Line 291

The bottom left figure shows an equation denoted as a red line.  However, the red line drawn is not as that in the equation.  The line drawn corresponds to perfect correlation between the two factors.  Although attempts are made to explain the figures (see lines 313-314), this should be made clear to avoid confusion.  This also applies to Figure 4 on page 10.

Page 9, Lines 303,308, and 326

The lines have been differentiated using a thick red line and a thin red line in both plots in Figures 3 and 4.

Point 11: Page 10, Line 310

Ensure superscript is used for ‘-2’ in ‘ms-2’.  There are many other occurrences of this.

Correction applied to the entire document.

Round 2

Reviewer 1 Report

The corrections that have been made in lines 86 to 89 of the revised manuscript show that the authors have misunderstood the comment that was made by this reviewer regarding the scaling factors that have to be used for assessing comfort effects and which must be set to kx=ky=kz=1 contrary to what is indicated. This raises serious doubts on the values of aWRMS that were computed for evaluating the comfort effects in this paper and on which the model relies for estimating discomfort.

The responses given by the authors to the comments that were made in the previous review do not convince this reviewer that this paper adds any significant value to making reliable decisions on sail or not-sail conditions in view of the poor correlation that was obtained with the proposed models. The fact that the measured acceleration data were collected for a purpose different to the objectives of this paper and for which some key information is missing certainly contributes to raising serious doubts on the value of the measured data, and consequently on the value of the proposed models.

Author Response

Response to Reviewer Comments

The response to comments is written in red starting with an indication of where reference page numbers and line numbers can be found in the revised manuscript.

Point 1: The corrections that have been made in lines 86 to 89 of the revised manuscript show that the authors have misunderstood the comment that was made by this reviewer regarding the scaling factors that have to be used for assessing comfort effects and which must be set to kx=ky=kz=1 contrary to what is indicated. This raises serious doubts on the values of aWRMS that were computed for evaluating the comfort effects in this paper and on which the model relies for estimating discomfort.

Page 2, lines 89 and 90

Amendments made in the manuscript in 'page 6 lines 217-220' and in the previous reply to the reviewer's previous comment show the accurate changes. This was clearly an error in writing and not a misunderstanding of the reviewer's comments. However, the changes have been made to lines 89-90 and show that kx=1.4, ky=1.4, and kz=1 were used for health assessments (MSI) as recommended in section 7.2.3 of the ISO, while the aWRMS model used (kx=1.0, ky=1.0, kz=1.0) from section 8.2.2.1 of the ISO.

Point 2: The responses given by the authors to the comments that were made in the previous review do not convince this reviewer that this paper adds any significant value to making reliable decisions on sail or not-sail conditions in view of the poor correlation that was obtained with the proposed models. The fact that the measured acceleration data were collected for a purpose different to the objectives of this paper and for which some key information is missing certainly contributes to raising serious doubts on the value of the measured data, and consequently on the value of the proposed models.

Page 1 lines 20-24, page 12 lines 393-397

The authors appreciate that the use of secondary data and the model results raise concerns about making reliable sailing decisions for this reviewer. However, the data used was from a similar project of which one of the authors was part. Key information was derived from the publication of that project, as referenced in this manuscript, and from informal requests by the authors. Where information is missing is due to restrictions. For instance, information on calibration could not be made available as the VMMS is a commercial product currently marketed by the developing company.

Amendments were made to clearly state that the model's in their current state do not make reliable sailing decisions but can, once the limitations highlighted have been addressed in future work. Therefore, the model results in themselves are significant findings in the use of machine learning for human response to vessel motions with recommendations given in this manuscript for further work.

Reviewer 2 Report

All comments have been addressed in revised manuscript

Author Response

Response to Reviewer Comment

The authors greatly appreciate the constructive, detailed, and informative comments given by this reviewer.